# Prospects for Biological Control of Macadamia Felted Coccid in Hawaii with *Metaphycus macadamiae* Polaszek & Noyes, a New Encyrtid Wasp Native to New South Wales, Australia

**DOI:** 10.3390/insects14100793

**Published:** 2023-09-29

**Authors:** Juliana A. Yalemar, Amber P. Tateno-Bisel, Stacey G. Chun, Mohsen M. Ramadan

**Affiliations:** 1Plant Pest Control Branch, Division of Plant Industry, Hawaii Department of Agriculture, 1428 South King Street, Honolulu, HI 96814, USA; juliana.a.yalemar@hawaii.gov (J.A.Y.); stacey.g.chun@hawaii.gov (S.G.C.); 2CDFA Fruit Fly Rearing Facility, 41-650 Waikupanaha Street, Waimanalo, HI 96795, USA; amber.tateno@cdfa.ca.gov

**Keywords:** macadamia, Eriococcidae, Encyrtidae, Hawaii

## Abstract

**Simple Summary:**

The macadamia felted coccid (MFC), *Acanthococcus* (=*Eriococcus*) *ironsidei* (Williams) (Hemiptera: Eriococcidae), is an invasive pest that has devastating impacts on 
the macadamia nut tree, in Hawaii and South Africa. MFC is a scale insect 
native to Australia where it has been recorded from smooth- and rough-shelled 
macadamia variants. Feeding causes discoloration and distortion of plant 
foliage, premature flower and nut drop, branch die back, and substantial 
reduction in nut production. Heavy infestations cause severe damage and death 
to large portions of trees. A survey conducted by Hawaii Department of 
Agriculture in New South Wales (NSW), Australia, found the undescribed 
endoparasitoid *Metaphycus* species is an important biotic factor for MFC. 
The parasitoid was imported to Hawaii for host specificity tests using closely 
related hemipterans, scale insect species, and other species of importance in 
Hawaii. Results indicated that this parasitoid is monospecific to MFC. This promising natural enemy was described as the new species, *Metaphycus macadamiae* Polaszek & Noyes sp. n (Hymenoptera: Encyrtidae). Laboratory parasitism averaged 30.2%, and parasitoids can feed on their hosts. Field parasitism in Australia is 32.7%. Several coccinellid predators and the aphelinid parasitoid, *Encarsia lounsburyi*, are local natural enemies of MFC in Hawaii, but their impacts alone are insufficient to suppress MFC populations. Introduction of biological control by release of *M. macadamiae* is expected to result in an effective long-term, sustainable solution for controlling MFC on macadamia nut trees in Hawaii or other infested areas in 
South Africa.

**Abstract:**

Macadamia felted coccid (MFC), *Acanthococcus ironsidei* (Williams) (Hemiptera: Eriococcidae), was first discovered in 2005 on the Island of Hawaii. Host plants are restricted to *Macadamia* species, with *Macadamia integrifolia* Maiden & Betche (Proteaceae) being grown in Hawaii for nut production. Approximately 6839 hectares macadamia nuts are harvested in Hawaii with an estimated farm value of USD 48.8 million (2019–2020 records). Exploration in Australia started in November 2013 for the evaluation of potential parasitoids being host specific for introduction into Hawaii. A dominant solitary endoparasitoid of MFC from New South Wales was discovered and described as *Metaphycus macadamiae* Polaszek & Noyes sp. n (Hymenoptera: Encyrtidae: Encyrtinae). Biology and host specificity testing were conducted at the Hawaii Department of Agriculture, Insect Containment Facility, on nine hemipteran and three lepidopteran eggs. Results indicated that *M. macadamiae* is host specific to MFC. There has been no evidence of parasitism or host feeding on any of the non-target insect hosts that were tested. Parasitoid emergence from the control (MFC) averaged 30.2% compared to 0% on non-target hosts. A low rate of parasitoid emergence in the laboratory (average 30.2%) and an increased rate of MFC nymphal mortality was due to adult feeding. Field parasitism reached up to 32.7% emergence in Alstonville, New South Wales, Australia. We report on the parasitoid performance in native Australia, rearing biology, host specificity testing, and the extant natural enemies associated with MFC in Hawaii. A petition to release this parasitoid for the biocontrol of MFC in Hawaii is pending. Once permitted for release, the colony will be shared with South African Mac Nut Association for their biocontrol program of this invasive pest. They will conduct their own testing before approval for release.

## 1. Introduction

The *Macadamia* genus is native to Australia and has been used for the commercial production of macadamia nuts in Hawaii for more than 80 years [1]. Two species of *Macadamia* have been used, primarily the smooth-shelled variety, *M. integrifolia* Maiden & Betche, and the rough-shelled variety, *M. tetraphylla* L.A.S. Johnson (Proteaceae) [2].

Commercial production of macadamia in Hawaii has grown exponentially to be the third most valuable crop, ranked as a commodity, in Hawaii after production of coffee, *Coffea arabica* L. (Rubiaceae), and seed corn, *Zea mays* L. (Poaceae) [3]. Typically, approximately 6475 hectares of macadamia nuts are harvested annually on the Island of Hawaii with an estimated farm value for the 2017–2018 crop of USD 53.9 million [4].

Macadamia felted coccid was first intercepted in 1954 on macadamia species imported into Hawaii [5]. Establishment did not occur until MFC was found infesting macadamia trees in South Kona, island of Hawaii, in February 2005 [6]. The MFC is a native Australian insect with plant hosts restricted to *Macadamia* varieties used for commercial production [2,5]. It infests all above-ground parts of trees and causes yellow spots on the leaves, die back on young seedling, and reduction in nut production [7,8,9]. Heavy infestation causes severe damage and eventual death of affected trees [10] (Figure 1). 

Macadamia felted coccid was established and has already spread across the island of Hawaii. In 2023, a survey was conducted by the Hawaii Department of Agriculture, and no infestation was found on the other major Hawaiian Islands (Kauai, Maui, and Oahu Islands) (HDOA-PPC, survey reports, 2023). Natural dispersal rates of this pest are very low, and distribution tends to occur primarily within infested trees [11]. In 2014, estimates on one farm indicated that as much as half a million pounds of wet in-shell macadamia nuts were lost because of MFC. According to USDA National Agricultural Statistics Service, the highest quantity of macadamia nuts, i.e., 6879.6 hectares, is harvested in Hawaii, and the farm value for the 2017–2018 crop was estimated at USD 53.9 million [4,12,13]. 

MFC belongs to the Family Eriococcidae with members that resemble mealybugs. The adult female is white to yellow brown and averages 0.7 × 1.0 mm in size [9], Figure 2A,B. Adult females are immobile and lay their eggs within felted sacs that are enclosed in their abdomens. A female lays 18–97 eggs during her lifetime of ≥50 days [14,15]. When the eggs hatch, the tiny crawlers move about, thus, spreading by wind or by hitchhiking. Long distance dispersal is mainly by passive transport of infested propagative material such as grafting budwood, scion wood cuttings, and potted nursery trees [9]. The life cycle takes six weeks in the summer (23.8–29.4 °C), and many overlapping generations are produced [16]. The female feeds by inserting her needle-like mouth parts into plant tissues and ingesting the sap (Figure 2B). Adult males are smaller in size, have wings, and do not feed, and their sole purpose is to mate with the females (Figure 2C).

In November 2013, the Hawaii Department of Agriculture, Plant Pest Control Branch (HDOA-PPC) initiated a foreign exploration in Australia to search for natural enemies of MFC. The host plants, *Macadamia* species, are native to Australia; therefore, it was the most likely place for potential natural enemies to be located. The HDOA-PPC believed that classical biological control may offer a long-term option for suppression of MFC. An encyrtid wasp, *Metaphycus* sp. (Hymenoptera: Encyrtidae), was collected as the dominant parasitoid and shipped to Hawaii in November 2013, propagated, and evaluated in the HDOA Insect Containment Facility (ICF). No other parasitoids emerged from this Australian collection. Many species of the genus *Metaphycus* have been used successfully in biological control programs against hemipteran pests with some great success in controlling the scale insects [17,18]; therefore, the unknown species of *Metaphycus* seemed like a potential biocontrol agent to control MFC. 

Preserved specimens were sent to Dr. Andrew Polaszek and Dr. John S. Noyes of the Natural History of Museum, London, United Kingdom, for description of this new species [19]. The wasp was described in 2020 and named as the new species, *Metaphycus macadamiae* Polaszek & Noyes sp. N. (Hymenoptera: Encyrtidae: Encyrtinae), a tiny solitary endoparasitoid. The female is light yellowish in color and is about 0.8 mm in length (Figure 3A,B). The male is dark in color and is approximately 0.6 mm in length (Figure 3D). The female lays a single egg inside each mature female host where it hatches, and the larva grows and develops, thus, killing the host in the process (Figure 2D). Females also host feed on MFC immatures [19]. No information was available on the host range of *M. macadamiae* in the scientific literature because it was a species not known to science. Host information of *M. macadamiae* from Australia is provided only by collections from *A. ironsidei* on *M. integrifolia* during the 2013 HDOA survey for MFC natural enemies.

In order to evaluate *M. macadamiae* as a prospective agent for the biocontrol of MFC in Hawaii, the life history, longevity, and fecundity was studied since this is a newly described species. Also host specificity testing was conducted to determine the host range of *M. macadamiae* and identify potential non-target insect hosts closely related to MFC. The objective was to determine whether *M. macadamiae* would have any negative impact on non-target insects in Hawaii either by feeding and or by ovipositing in the absence of its natural host MFC. Here, we report these findings: parasitoid performance in the native region, and extant natural enemies of MFC in Hawaii. Also, for permission purposes from state officials and USDA-APHIS, the Environmental Assessment for *M. macadamiae* has been drafted and is currently under evaluation. 

## 2. Material and Methods

### 2.1. Insect Containment Facility Settings and Rearing Conditions

All biology and host specificity testing for *M. macadamiae* was conducted in the HDOA Insect Containment Facility. Wasps were reared for 10 generations before testing began. The Insect Containment Facility was 22.0 ± 1.0 °C at night and 34.0 ± 2 °C during the day, with 60–80% RH and 13L: 11D photoperiod. The purpose was to determine if this parasitoid would have any negative impact on other non-target insects in Hawaii.

Older nuts of macadamia were collected from the field and propagated in 2 L black plastic pots with drainage holes and saucer (17 cm top Ø, 12 cm base Ø, 13 cm height). Pots were held under 75% shade, and seeds germinated within 6–7 weeks. Seedlings were held until they acquired their 3rd or 4th set of leaves, before they were ready to be exposed to MFC infestation. Infested macadamia branches were brought back from the field (South Kona, 19°08′06.64″ N, 155°50′39.5″ W, 516 m) and placed on pots between seedlings. Infested seedlings took 7–8 weeks to obtain enough MFC infestation after which they were exposed to the parasitoid. Two to three pots of infested macadamia seedlings were placed in collapsible lightweight aluminum cages (30 × 30 × 60 cm) with clear vinyl doors and 70 mesh chiffon covered rear and top sides. 

### 2.2. Host and Parasitoid Rearing

Initial parasitoid cohorts originated from infested macadamia from New South Wales, Australia. Founder cohorts were 55 wasps established in the Containment Facility, Honolulu, Oahu Island (21°17′56.00″ N, 157°50′19.69″ W, 6 m). Twenty newly emerged females and ten males were released in each cage for oviposition. A few drops of honey were smeared on the top and sides of each cage for adult feeding. Water was provided in a cup (Deli container with lid, and a cotton wick, 470 mL). After 15 days, newly emerged parasitoids were collected and used for exposure to new MFC-infested seedlings. Parasitoids were reared continuously in the HDOA Insect Containment Facility that was 22.0 ± 1.0 °C at night and 34.0 ± 2 °C during the day, with 60–80% RH and 13L: 11D photoperiod, under fluorescent light plus natural sunlight through window glass panels to facilitate mating of parasitoids [20]. The colony was reared for 10 generations before host testing was conducted. 

### 2.3. Life History, Longevity, and Fecundity of M. macadamiae

#### 2.3.1. Life History

Oviposition was examined by placing excised MFC-infested macadamia foliage inside a Petri dish (14.5 cm Ø × 2.0 cm height plastic Petri dishes) with *M. macadamiae* adults and observing them under a dissecting microscope (Trinocular Stereo Microscope with top and bottom lights). Wet filter paper was added to keep the leaf moist. Information on the duration of the life cycle was determined by keeping track of the first day of exposure to MFC and the first day of parasitoid emergence. 

#### 2.3.2. Longevity

Adult longevity was determined by collecting newly emerged parasitoids and placing them in 10 mL vials. Vial covers were modified to include a 5 mm Ø hole in the center, covered with a fine mesh cloth. Honey was dotted on the cloth as food. Parasitoids were examined daily for mortality. A total of 20 males and 25 females were collected and held in such vials at 5 parasitoids per vial separated by sex. 

#### 2.3.3. Fecundity

Fecundity studies were based on the female potential fecundity, potential reproductive output of an individual female over its lifetime. Newly emerged females were collected and held individually in 10 mL vials and were fed honey. Ten females at each desired age (<1–5-week-old) were dissected in saline solution, and their mature eggs were counted. Mature ovarian eggs are recognized as characteristic ovarian encyrtid eggs with a double-bodied shape, consisting of two ovoid bulbs connected by a narrow tube [21] (Figure 4). The counts reflect the potential fecundity because they are mature ovarian eggs that parasitoids can produce. 

### 2.4. Host Specificity Testing

Two genera in the family Eriococcidae, both adventives, are listed in the Hawaiian Terrestrial Arthropod Checklist that has all adventive members [22]. The genus *Acanthococcus* in Hawaii has one listed member, *Acanthococcus araucariae* (Maskell), a pest found on needles of *Araucaria* spp. (Araucariaceae). The second genus, *Eriococcus,* has only one listed species, *Eriococcus coccineus* (Cockerell), now moved to *Acanthococcus coccineus* (Cockerell), a new name by Miller and Gimpel 2000 [10], which is a pest on cactus (Cactaceae) that was reported only in Kauai Island. A recent addition to the eriococcid family in Hawaii is *Tectococcus ovatus* Hempel, a weed biological control agent released in 2012 to control Strawberry Guava, *Psidium cattleianum* Sabine, (Myrtaceae), Table 1, and is included in host specificity testing.

A total of twelve insect species were tested against *M. macadamiae* of which nine are economically important and endemic members of the Hemiptera: Sternorrhyncha (e.g., Aleyrodidae, Coccidae, Dactylopiidae, Eriococcidae, Halimococcidae, Pseudococcidae, and Triozidae). Some members of these families are reported hosts of *Metaphycus* spp. [23]. Additionally, some encyrtids may attack lepidopteran eggs [24]; therefore, we included three representatives of the Order Lepidoptera in our tests, i.e., one endemic *Vanessa tameamea* (Eschscholtz), one naturalized nymphalid, *Danaus plexippus* (Linnaeus), and one beneficial arctiid moth, *Secusio extensa* (Butler), released in Hawaii to control the fireweed, *Senecio madgascariensis* Poir., and *Delairea odorata* Lem. (Table 1).

Infested branch cuttings collected from the field or infested plant seedlings reared at the HDOA Insectary containing non-target insects were exposed to *M. macadamiae* for host specificity testing. A plant or plant cutting infested with one of the non-targets was placed in a (30 × 30 × 60 cm) collapsible metal cage and exposed to naïve newly enclosed groups of ten females and five males of *M. macadamiae* in each cage and held until parasitoids died. In another cage, an MFC-infested macadamia seedling was placed, and the same number of parasitoids were released inside the cage. Host specificity evaluations were based on no-choice tests. All host plants contained all stages nymphs and pupae of non-target insects. After one month, non-target insects were dissected and examined for evidence of parasitism. Three infested leaves were randomly picked from each control, and the number of MFC on each leaf was tallied and examined for parasitism. Parasitism was determined by the presence of parasitoid circular exit holes (Figure 3C), unemerged parasitoid cadavers, and dead MFC due to parasitoid probing marks. Parasitism in the control replicates was determined with adult parasitoid emergence and parasitoid circular exit holes. In the case of the tested lepidopteran eggs, plant parts containing eggs were collected and placed in Petri dishes (2.0 cm height × 14.5 cm Ø). Five females and five males of *M. macadamiae* were released inside each Petri dish. Unhatched eggs were examined under a dissecting microscope for evidence of probing or oviposition which left traces of recognizable blacken melanized oviposition scars. All tests were replicated three times.

### 2.5. Parasitoid Field Performance in Australia 

*M. macadamiae* was dissected from MFC on infested leaves of *Macadamia integrifolia* collected in Alstonville, NSW, Australia. Two batches of infested macadamia leaves were collected from different trees that were not known to be sprayed with insecticides, and subsequently shipped to HDOA ICF. One consignment was obtained on 19 November 2013 (*n* = 150 infested leaves, 28°51′20.14″ S, 153°26′31.40″ E, 136 m), from Alstonville, NSW, Australia, and Department of Primary Industry of Australia and another batch of (*n* = 130) infested leaves, on November 25, 2013, from the same locality (28°49′13.51″ S, 153°23′44.65″ E, 168 m). MFC individuals collected on the infested macadamia leaves yielded only adult *M. macadamiae* wasps. Mean numbers of MFC/leaf, % parasitism by *M. macadamiae*, and % predations were recorded from leaves and petioles (*n* = 30) of infested leaves. Parasitism and predation rates were determined by shape of parasitoid exit holes or predation chewing holes. Dominant predators on the trees were photographed, and one species was identified using keys of Australian Lady Beetles [25]. Parasitism was determined by counting the MFC with circular holes of parasitoid emergence (Figure 3C, white arrows), and predation was determined by the larger oblong irregular holes on scales (Figure 3C). 

### 2.6. Extant Natural Enemies in Hawaii

Relative abundance of local natural enemies of MFC on three orchards on the island of Hawaii; Pahala, South Hilo (19°08′08.35″ N, 155°50′44.78″ W, 503 m); Honokaa, North Hawaii (20°04′5.07″ N, 155°28′19.92″ W, 476 m); and Honomalino, South Kona (19°08′06.64″ N, 155°50′39.5″ W, 516 m) were studied by counts of parasitoids and predators on sticky traps. Ten randomly selected infested trees per orchard were designated. Yellow sticky traps set for flying insects (5 cm wide × 15 cm length) with glue on one side were placed one per tree on infested branches at ≤2 m above ground. Traps were replaced every month, and numbers of parasitoids and predators stuck on traps were microscopically tallied as means ± SEM of parasitoids and five species of coccinellids/trap/month/orchard during twelve months of 2015. *Encarsia lounsburyi* (Berlese & Paoli) (Hymenoptera: Aphelinidae), a parasitoid of male MFC, and *Curinus coeruleus* Mulsant (Coleoptera: Coccinellidae) were the dominant parasitoid and ladybeetle in macadamia fields, respectively, during this survey [12]. 

### 2.7. Statistical Analysis and Vouchers

For studies on field parasitism in Australia, an analysis of variance was used to assess the potential significance of differences in the number of parasitoids produced by the *M. macadamiae* parasitism, % parasitism, and % predation. Means were separated with Tukey’s standardized range honestly significant difference test and unequal variances Welch’s *t*-test at *p* = 0.05 level [26]. Percentage data were transformed arcsine √ proportion before analysis. Voucher specimens and paratypes of *M. macadamiae* were placed in the insect reference collection of the HDOA, the Bernice P. Bishop Museum, Honolulu, Hawaii, and the Department Life Science Collections of the Natural History Museum, London. Vouchers specimens are deposited in Australian National Insect Collection, CSIRO, Canberra, Australia, United States National Museum, Washington D.C., USA, and the Hawaii Department of Agriculture insect collection [19]. 

## 3. Results 

### 3.1. Life History, Longevity, and Fecundity of M. macadamiae

#### 3.1.1. Life History and Longevity

Adult emergence ranged from 12–21 days after MFC exposure to parasitoids depending on temperature. In the summer, June–August, maximum indoor temperature ranged 31.9–32.2 °C, when days were warmer males were seen emerging as early as 12 days after exposure. *M. macadamiae* can have multiple generations per year under laboratory conditions. 

*M. macadamiae* females had on average a longer lifespan compared to the short-living males. Mean ± SEM of female longevity was 32.9 ± 3.1 d and that of male longevity was significantly shorter at 8.3 ± 1.4 days (*t* = 7.1679, df = 32.57, *p* < 0.0001). 

#### 3.1.2. Fecundity

Number of mature ovarian eggs in ≤1 week-old females seen in dissections ranged 3–10 eggs with a mean ± SEM of 5.3 ± 0.73 mature eggs. Maximum egg production peaked at 14 ovarian eggs in one-week-old females with a mean of 10.4 ± 0.8 mature eggs per female and declined thereafter to 7.8 ± 0.4 and 4.1 ± 0.2 as the female aged to 2–5-week-old (Figure 4). This indicated that *M. macadamiae* is a synovigenic species that produces mature eggs throughout its adult life and resorbs eggs at an older age (F_3,36_ = 1.823; *p* < 0.0001). 

### 3.2. Host Specificity Testing

Host specificity study proved that *M. macadamiae* is specific to MFC. There was no parasitoid emergence from any of the twelve non-targets tested. Moreover, dissections revealed no evidence of parasitism nor host feeding on non-targets. Although parasitism rates varied between controls, all had some degree of emerged parasitoids, ranging 6.7–62.5% (Table 1). 

### 3.3. Field Parasitism Evaluation of M. macadamiae in Australia

The rate of parasitism in the field ranged 21.2–32.7% in Alstonville, during the November 2013 sampling. Predation reached a mean of ≥5% with two recognized predators. Mean infestation ranged 16.7–19.9 MFC/infested leaf during November 2013. Parasitoids perform better on leaves, with a higher parasitism rate on leaves than petioles (Figure 5). 

### 3.4. Extant Natural Enemies of MFC in Hawaii

Several coccinellid predators and the aphelinid parasitoid, *Encarsia lounsburyi,* are known natural enemies of MFC in Hawaii, but their impacts alone are insufficient to suppress MFC populations. The average count of trap catches per month in three macadamia orchards on the island of Hawaii during 2015 showed that the parasitoid *E. lounsburyi* is thriving especially in Honokaa Orchards. In comparison, predation with coccinellid counts were significantly low (Figure 6). 

The Pahala site had no significant differences between the densities of *E. lounsburyi* and *C. coeruleus,* whereas Honomalino had mean count differences of 25 parasitoids to 5 predators, measured as individuals/trap/month. Honokaa had no incidence of *C. coeruleus,* and only *E. lounsburyi* was present with mean counts above 300 individuals/trap/month (Figure 6).

## 4. Discussion

Field surveys of natural enemies in macadamia nut orchards on the island of Hawaii revealed the presence of several predatory coccinellid beetles (*Halmus chalybeus* (Boisduval), *Rhyzobius forestieri* (Mulsant), *Sticholotis ruficeps* Weise, and *Telsimia nitida* Chapin), and the parasitic wasp *Encarsia lounsburyi*, associated with MFC [15]. The parasitic wasp, *E. lounsburyi,* was first found associated with MFC infestations on the island of Hawaii in 2005 [27]. *E. lounsburyi* is recorded to parasitize a range of diaspidid scale insects and is not host specific to MFC like the Australian *M. macadamiae* [28]. All species of Diaspididae are adventive in Hawaii [22].

Local natural enemies in Hawaii may play a minor part in reducing numbers of MFC, but do not prevent populations from reaching damagingly high densities. They are considered generalists, and their impacts on high MFC infestations are inadequate to reduce the pest populations below economically injurious levels. With the high MFC populations that are typical in Hawaii, macadamia nut orchards biocontrol is hopeless without an effective specialized agent [29]. Our laboratory studies show that MFC parasitism rate with *M. macadamiae* can range from 11 to 62% (Table 1), higher than estimates from field parasitism in Australia (21–33% parasitism). In addition, female wasps host-feed on MFC nymphs, thereby adding to the mortality of MFC attributable to the wasp. The addition of *M. macadamiae* to the assembly of natural enemies exploiting MFC will hopefully decrease pest population levels so densities of MFC are reduced to tolerable levels as in the native Australia.

*Metaphycus macadamiae* is a newly described species. No records of performance exist in the scientific literature. Other members of this genus are known as primary endoparasitoids of scale insects, and a few species of the Encyrtidae family parasitize eggs of some lepidopterans [30]. 

Historically, three species of *Metaphycus* have been released in Hawaii from previous biological control introductions, i.e., *Metaphycus clauseni* (Timberlake), *M. helvolus* (Compere), and *M. luteolus* (Timberlake), and were purposefully released as biological control agents of soft scale insects and mealybugs in the period 1934–1964. The parasitoids originated from California, but the establishment is unknown for the three species [22,31]. Additionally, seven *Metaphycus* species were accidently introduced to Hawaii and recorded as natural enemies of scale insect pests on major Hawaiian Islands: *M. alberti* (Howard), *M. anneckei* Guerrieri & Noyes, *M. claviger* (Timberlake), *M. eruptor* (Howard), *M. flavus* (Howard), *M. portoricensis* (Dozier), and *M. stanleyi* Compere. All seven species were established on various islands [22,31]. No non-target parasitism of native insects in Hawaii was recorded by *Metaphycus* species since their first introduction in 1934 [31]. 

In 2017, MFC was found severely infesting macadamia nut trees in Barberton valley, Mpumalanga province, South Africa [32,33], where it is also a devastating pest to the country’s macadamia nut producing industry, due to the rapid spread within a month to White River Macadamia orchards, presumably through infested plant material [34]. South Africa has been the world’s largest producer of the macadamia since the 2010s [35,36].

A starter colony of *M. macadamiae* (200 wasps) was hand-carried from Hawaii to a South African quarantine facility for propagation The colony arrived in good condition (Dr. Mark Wright, University of Hawaii at Manoa, unpublished). Unfortunately, this culture was lost before any permission to release in the field. In addition to our information on host testing, South Africa may need to consider study specificity on other introduced eriococcid species of South Africa (i.e., *E. coccineus* Cockerell, *E. araucariae* Maskell, *E. leptospermi* (Maskell), and the native species *Calycicoccus merwei* Brain [37]. 

In Australia, MFC is only a problem in newly infested localities until natural enemies catch up to exert adequate control [9]. The infestation and rate of parasitism from the sampled leaves indicated that *M. macadamiae* is the dominant natural enemy. *M. macadamiae* seems to perform better on leaves showing higher rates of parasitism than on petioles and stems. Infested leaves may be more attractive to the parasitoids because of higher MFC density and presence of honeydew drops on leaves (Figure 5). 

The longevity of the adults of *Metaphycus* species is known to be influenced by available food sources of insect hosts and sucrose availability of adult diet. Non-fed adults did not survive past the second day, while honey-fed individuals lived more than six days in some species [38]. In our laboratory experiment, the MFC-infested leaves would provide a sugar diet content to the parasitoid. This may also happen in the field without dependence on flowering plants for nectar (Figure 2C). Still, female *M. macadamiae* are observed to partake in insect host feeding. They were observed to penetrate the young hosts by ovipositor and then to feed on oozing fluid from immature MFC. Females do not use these young, shriveled individuals for oviposition. Host feeding increases parasitoid longevity and their potential fecundity in naïve wasps [38]. Similar results of longevity of *Metacphycus* spp. indicate non-ovipositing females lived on average about eight days, whereas ovipositing females lived on average about five days [39]. 

Insect growth regulators (IGR) and horticultural oils insecticides are currently used to help control MFC in Hawaii. Some of these insecticides have negative impacts on natural enemies that are present in orchards, such as Coccinellidae and parasitoids. Based on studies by Gutierrez-Coarite et al. 2017 [15], insecticide treatments with IGR compounds are appropriate when MFC populations are high, whereas horticultural oils combined with natural enemies are most effective when populations are low. The use of chemicals to control MFC may negatively impact the extant natural enemies in macadamia orchards and honeybees. Honeybees are essential for pollination in macadamia fields. Therefore, a specific parasitoid like *M. macadamiae* is desirable. 

Cultural control is an essential component of the integrated pest management system utilized by macadamia orchards in Hawaii. Keeping macadamia nut trees healthy by maintaining the right soil moisture, fertilizing, cleaning, and pruning helps defend them against MFC infestation. In addition, MFC infestation is heaviest in shaded sites within the orchards and on trees with sucker shoots [7,28]; hence, trees need to be pruned and cleaned regularly to prevent build-up of MFC populations. However, cultural management alone is insufficient to reduce MFC impacts. 

## 5. Conclusions

Like several other *Metaphycus* species released in previous biological control programs, this endoparasite wasp is highly unlikely to produce any adverse effect on the fauna of Hawaii, both native and beneficial introduced species including the lepidopteran eggs. Being a monophagous species, *M. macadamiae* females will exclusively seek MFC nymphs for host feeding, and young adult MFC for oviposition. MFC and macadamia crops are expected to be the only organism in Hawaii that will be affected by the release of this parasitoid. Observations in the native region indicate that *M. macadamiae* is restricted to MFC hosts reaching up to 33% parasitism in Australian fields, making MFC a minor pest of *Macadamia.* Host specificity testing in the HDOA-PPC Insect Containment Facility demonstrated that *M. macadamiae* will only attack the target pest *A. ironsidei*. There are few immigrant pest species closely related to MFC existing in Hawaii, further minimizing any potential for unexpected impacts in Hawaii once *M. macadamiae* is released. This report of host specificity and parasitoid performance should support decision making for release permits in Hawaii or infested regions in South Africa. The parasitoid was unable to successfully emerge or feed on any tested non-target species.

## Figures and Tables

**Figure 1 insects-14-00793-f001:**
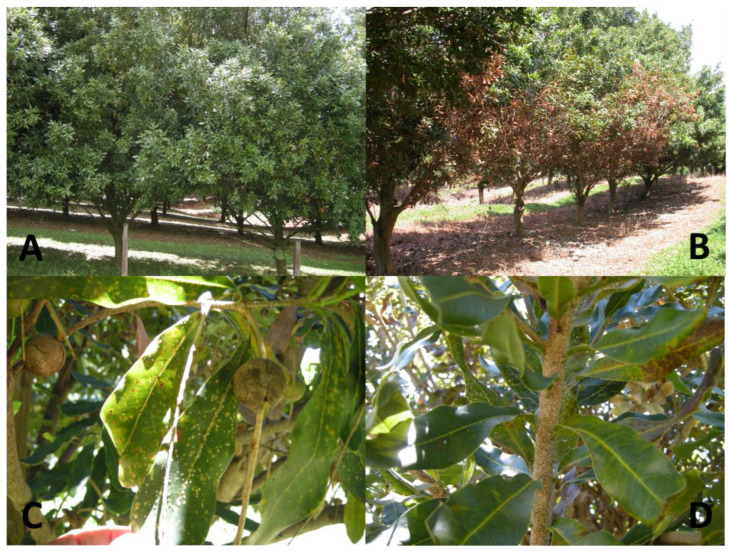
(**A**) Australian macadamia orchard (NSW), in November 2013, with MFC as a minor pest, (**B**) Hawaii macadamia orchard 2005 severe infestation, (**C**) MFC infestation on leaves and nuts in Hawaii, and (**D**) infestation of stems in Hawaii.

**Figure 2 insects-14-00793-f002:**
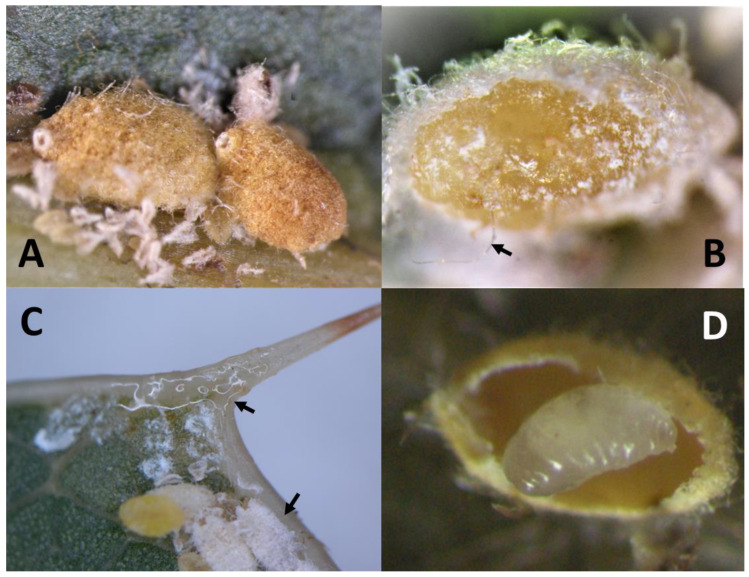
(**A**) MFC mature female, dirty white to pale yellow, scale about 1.5 mm in length, with a raised circular opening at the posterior end, (**B**) female MFC orange in color, showing long stylet mouth parts (arrow), (**C**) honeydew produced on leaves by the nymphs of MFC (arrows honeydew and male white nymphs), and (**D**) *Metaphychus macadamiae* larval stage dissected from female MFC.

**Figure 3 insects-14-00793-f003:**
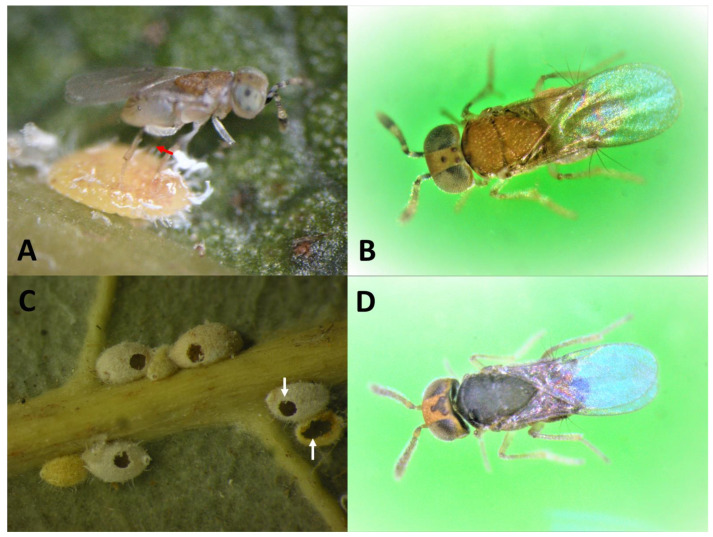
(**A**) Female *M. macadamiae* probing the host for oviposition (arrow showing ovipositor), (**B**) habitus of female *M. macadamiae*, 0.63–0.78 mm in length, (**C**) MFC with circular parasitoid exit holes (upper arrow) versus predation chewing holes (lower arrows), and (**D**) habitus of darker male *M. macadamiae* smaller in size, 0.46–0.66 mm in length.

**Figure 4 insects-14-00793-f004:**
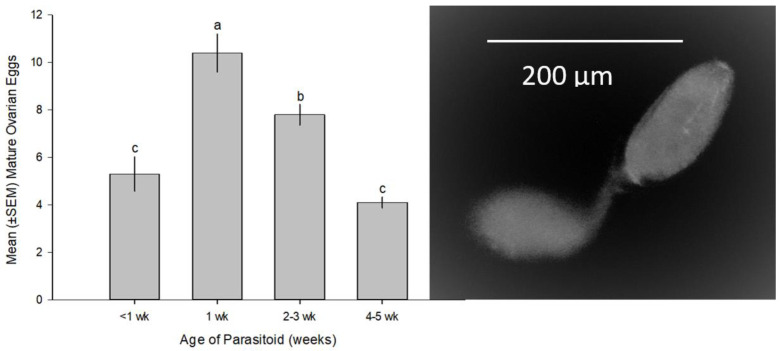
*M. macadamiae* potential fecundity, and ovarian maturation peaked at one-week old female. The image is a mature stalked egg typical of encyrtiform ovarian egg and is two-bodied, with a stalk tube between the two bulbs. Comparisons were performed using Tukey–Kramer HSD (ANOVA: F_3,36_ = 21.8233, *p* < 0.0001). Bars topped by different letters, are significantly different.

**Figure 5 insects-14-00793-f005:**
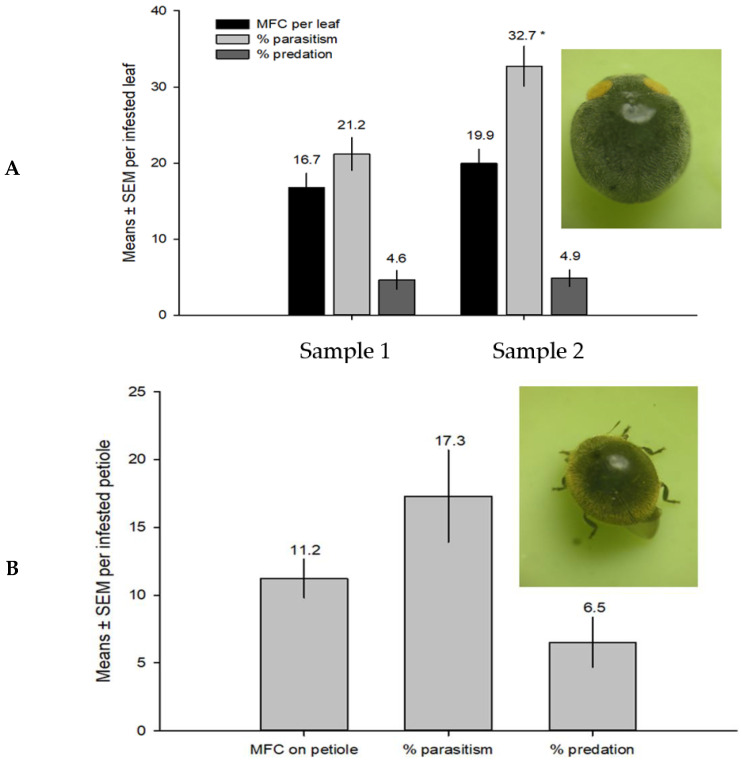
Field infestation by MFC, and *M. macadamiae* performance on (**A**) leaves and (**B**) on petioles parasitism and % predation, during November 2013, at Alstonville, NSW, Australia. Two dominant predators, with only one species identified, and inset bottom is *Rhyzobius ventralis* (Erichson). (*) significantly different (*p* < 0.05).

**Figure 6 insects-14-00793-f006:**
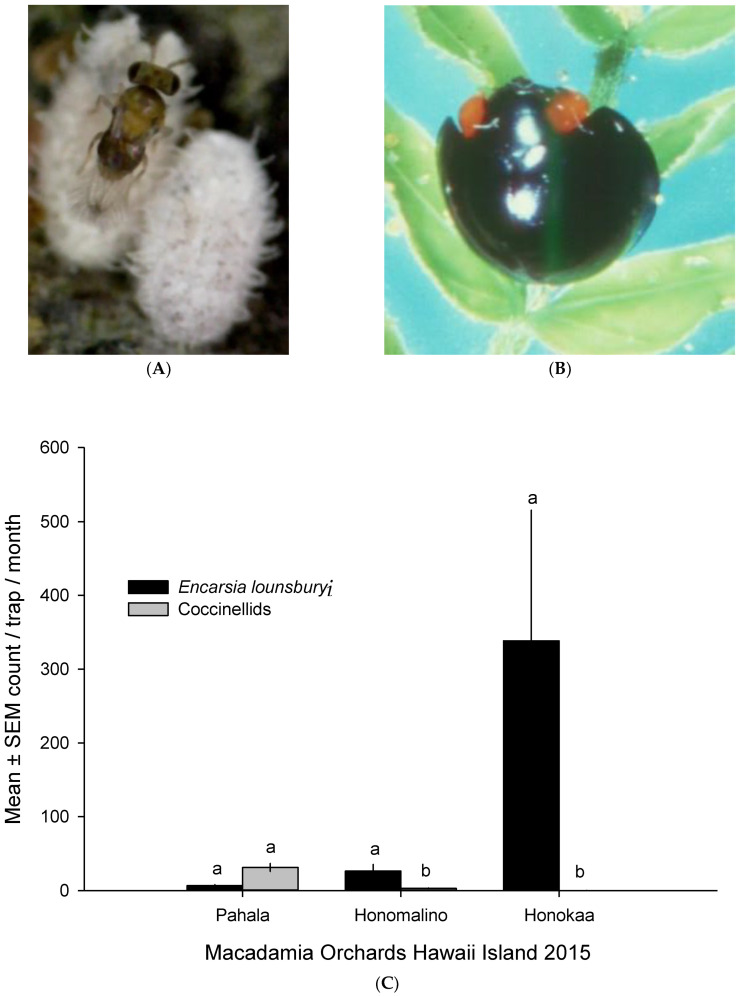
Relative abundance of extant natural enemies of MFC on three orchards on the island of Hawaii during 2015. Values are means ± SEM of parasitoids and five coccinellids per sticky trap per month placed on infested macadamia trees. (**A**) *Encarsia lounsburyi* (Hymenoptera: Aphelinidae) on MFC male scale. White male scale is about 1.0 mm long. (**B**) *Curinus coeruleus* (Coleoptera: Coccinellidae), a dominant lady beetle on macadamia fields. (**C**) Parasitoid or predator bars topped with same letters in three orchards are not significantly different (*p* > 0.05).

**Table 1 insects-14-00793-t001:** *Metaphycus macadamiae* (Hymenoptera: Encyrtidae) host specificity study with non-target insects and macadamia felted coccid (MFC), *Acanthococcus ironsidei*, as the control.

Non-Target Insect Hosts and Host Plants					Mean% Parasitism Based on *Metaphycus emergence* and Non-Target Dissections (*n* = 3)		
Scientific Name, Order, and Family	Stage Tested	Status	Source	Host Plant and Infested Plant Part Used	MFC (Control) % *M. macadamiae* Emergence	Total Non-Target Insects Dissected	Non-Target Insects % Parasitism
*Tectococcus ovatus* (Hempel) Hemiptera: Eriococcidae	Adult and Nymph	Biocontrol agent	Lab-reared HDOA, Oahu	*Psidium cattleianum* Whole plants and seedlings	6.7	200	0
*Acanthococcus araucariae* (Maskell) Hemiptera: Eriococcidae	Adult and Nymph	Immigrant	Field collected, Molokai	*Araucaria* sp. Cuttings	17.5	420	0
*Thysanococcus pandani* (Stickney) Hemiptera: Halimococcidae	Adult and Nymph	Immigrant	Field collected, Maui	*Pandanus tectorius* Whole plants	21.7	300	0
*Colobopyga pritchardiae* (Stickney) Hemiptera: Halimococcidae	Adult and Nymph	Endemic	Field collected, Hawaii	*Pritchardia* sp. Cuttings	15.2	500	0
*Dactylopius opuntiae* (Cockerell) Hemiptera: Dactylopiidae	Adult and Nymph	Biocontrol agent	Field collected, Oahu	*Opuntia ficus-indica* Cuttings	11.7	1100	0
*Saissetia oleae* (Oliver) Hemiptera: Coccidae	Adult and Nymph	Immigrant	Lab-reared HDOA, Oahu	*Erythrina variegata* Whole plants and seedlings	30.6	1642	0
*Pseudococcid montanus* (Erhorn) Hemiptera: Pseudococcidae	Nymphs and pupae	Endemic	Field collected, Oahu	*Freycetia arborea* Cuttings	62.5	354	0
*Pariaconus ohiacola* (Crawford) Hemiptera: Triozidae	Nymphs and pupae	Endemic	Field collected, Oahu	*Metrosideros* sp. Cuttings	54.3	300	0
*Tetraleurodes acaciae* (Quaintance) Hemiptera: Aleyrodidae	Egg	Immigrant	Field collected, Oahu	*Erythrina variegata*, Seedling	38.3	300	0
*Vanessa tameamea* (Eschscholtz) Lepidoptera: Nymphalidae	Egg	Endemic	Lab-reared PEPS, UHM	*Pipturus albidus* Eggs placed on filter paper	57.3	55	0
*Danaus plexippus* (L.) Lepidoptera: Nymphalidae	Egg	Naturalized	Field collected, Oahu	*Colotropis gigantea * Foliage	19.4	30	0
*Secusio extensa* (Butler) Lepidoptera: Erebidae	Egg	Biocontrol agent	Lab-reared HDOA, Oahu	*Senecio madgascariensis* Foliage			

## Data Availability

The data presented in this study are available on request from the corresponding author.

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
