# Peer review of "Prospects for Biological Control of Macadamia Felted Coccid in Hawaii with Metaphycus macadamiae Polaszek & Noyes, a New Encyrtid Wasp Native to New South Wales, Australia"

_insects, 2023, doi:10.3390/insects14100793_

Round 1
Reviewer 1 Report
Line 36. Chemical control should be specified in the introduction.
Line 73. Labels on pictures of Figure 1 are not clear
Line 91. Update version of farm value
Line 94. Felted sacs enclose their whole body not only their abdomen
Line 327. Figure 5. parasitism % should be compared with more than one predator species since ventralis could predate MFC but can have other preferences such as aphids.
Comments
Why field parasitism and predation percentages on MFC were not conducted for Hawaii as were for Australia? It is important to know these data before releasing metaphycus to compare how parasitism increased after and thereafter MFC abundance in macadamia orchards. If you don't have this data I suggest adding information from other references about it to the discussion.
An alphabetical order of references is needed
Author Response
Many thanks to reviewer 1 for the suggested notes. Please see the attached file for responses to Reviewer 1. A final version of the manuscript will be edited by a native English speaker.
Comments and Suggestions for Authors
Thank you for reviewer 1 for reading and editing our manuscript. All suggested editing are considered in the final resubmitted manuscript.
Line 36. Chemical control should be specified in the introduction.
Response: Thank you, statement of chemical control removed from abstract. Added to discussion section.
Line 73. Labels on pictures of Figure 1 are not clear.
Response: Labels in figure 1 changed to white for clarity.
Line 91. Update version of farm value
Response: Stated in manuscript that harvested farm value fluctuates in Hawaii and the highest value for the 2017-2018 crop is estimated at $53.9 million [4,13].
Line 94. Felted sacs enclose their whole body not only their abdomen.
Response: This referring to eggs in the abdomen. Adult females are immobile and lay their eggs within felted sacs that enclosed their abdomens.
Line 327. Figure 5. parasitism % should be compared with more than one predator species since ventralis could predate MFC but can have other preferences such as aphids.
Response: Thank you, we estimated predation by counting MFC with chewed predation holes. Two dominant predators were encountered, only one was identified as Rhyzobius ventralis (Erichson).
Comments
Why field parasitism and predation percentages on MFC were not conducted for Hawaii as were for Australia? It is important to know these data before releasing metaphycus to compare how parasitism increased after and thereafter MFC abundance in macadamia orchards. If you don't have this data I suggest adding information from other references about it to the discussion.
Response: The parasitoid has not been permitted for release in Hawaii yet. Data on rates of infestation and parasitism before and after the release will be determined and monitoring of parasitoid performance. In discussion we mentioned that recent parasitism and predation are not adequate for MFC control on Hawaii Island.
An alphabetical order of references is needed.
Response: References adjusted to preference of INSECTS Journal; numerical order is the sorting method for references in this journal.
Reviewer 2 Report
The manuscript is an important documentation of the early steps of classical biological control of macadamia felted coccid, an introduced species in HI. The methods and the work are sound, but the English is full of small annoying errors and needs to be edited by a native speaker.
See the attached Word Doc for my detailed comments

The English is not up to the standard for publication. See my attached Word document for examples. It needs to be professionally edited before being accepted.
Author Response
Thank you so much for reading and editing our manuscript. Hope the responses satisfy your concerns. A final revised manuscript will be resubmitted.
The manuscript is an important documentation of the early steps of classical biological control of macadamia felted coccid, an introduced species in HI. The methods and the work are sound, but the English is full of small annoying errors and needs to be edited by a native speaker.
See the attached Word Doc for my detailed comments peer-review-31962161.v1.pdf
Response: Thank you very much for reading our manuscript, responses to your concerns improved our manuscript.
Comments on the Quality of English Language
The English is not up to the standard for publication. See my attached Word document for examples. It needs to be professionally edited before being accepted.
Response: minor editing of English was conducted. A revised manuscript will be edited by a native English speaker colleague before resubmission.
General Comments
The manuscript, while more or less understandable, strongly needs editing by a native English speaker to improve wording and grammar. Currently, it reads very roughly and contains many errors, throughout the whole manuscript, ranging from just annoying to grammatically wrong. A few of these problems are listed below as examples. No effort is made here to list them all.
Response: Thank you for reading our manuscript, the revised manuscript will be edited by a native English speaker before resubmission. All suggested changes are considered.
Abstract
Line 18 insert “and” between “trees” and “feeds”
Response: corrected but several sentenced removed from this simple summary
The wording in the Simple Summary implies host range test was done in NSW, (lines 22 ff) but the Abstract states it was done in HI (line 41 ff).
Response: Thank you, for clarity, in simple summary authors stated, “Australia found that the undescribed endoparasitoid Metaphycus species is an important biotic factor of MFC”. The words “host specific” removed for clarity. This was introduced to Hawaii for host testing and description.
Introduction
Line 56 change genera to genus
Response: Thank you, changed to genus.
Line 60 change 1930’s to 1930s
Response: Thank you, changed to 1930s
Line 110 inert semicolon (; ) between Australia and therefore
Response: Thank you a (;) inserted.
Line 118, insert semicolon (; ) between [17,18] and therefore, not a comma
Response: Thank you a (;) inserted.
Line 122 change identified to described; only species that already have names can be identified
Response: Thank you for clarification, changes made.
Line 127 State what life stage of the scale it is that the parasitoid inserts its eggs into
Response: The parasitoid usually prefers mature female scale for oviposition. Changed to mature female in text for clarity.
Line 139 Change identified to described
Response: Thank you, done in all manuscript.
Link 146 Drop to be approved
Response: Thank you “be approved” dropped.
Methods
Line 157 add seeds between and and germinated
Response: Thank you “seeds’ added.
Line 165, since whole line is in italics, parasitoid name should NOT be in italics
Response: Thank you the parasitoid name dropped.
Line 166 insert parasitoid between initial and cohorts
Response: Thank you, done.
Line 170, what second word? (in name of honey)? I am guessing you mean Premium
Response: The whole line of honey name was dropped by another reviewer.
Line 204, what are “pulps”? (in caption for Figure 4)
Response: Thank you for correcting our mistake. We meant bulbs.
Line 215 the phrase “has been added” makes it sound like this manuscript was revised by adding that species to your test list. Saying that at this point seems odd.
Response: rewording the text for clarity, changed to is included
Line 233 the combination of “all various” is not correct English
Response: Thank you for the word “various” removed from text.
Line 279, figure 6 should be Figure 6
Response: Thank you, changed to Figure 6.
Line 283 By “predation” I assume you mean “host-feeding.” If so, say it that way to avoid confusion with predators.
Response: Reworded in text: For studies on field parasitism in Australia, an analysis of variance was used to assess the potential significance of differences in the number of parasitoids produced by the M. macadamiae parasitism, % parasitism, and % predation.
Results
Line 296 “parasitoid exposure to MFC” is conceptually backwards, as it is the scale that is exposed to the parasitoid.
Response: Thank you, this changed for clarity.
Line 305-312 This section is not clear. Are your data the number of eggs in the ovarioles as seen in dissection or are they the number of eggs laid per female when females are exposed to suitable hosts? The text reads sort of both ways at different points. To me, the term “mature ovarian eggs” implies eggs still in the ovary, just mature.
Response: This part describes the potential fecundity of the parasitoid. Changed for clarity to “Number of mature ovarian eggs in ≤1 week-old females seen in dissections ranged 3-10 eggs with a mean ± SEM 5.3 ± 0.73 mature eggs”.
Line 323 In discussion predation in Australia a predation rate is given. How was that measured? I do not remember seeing any description of such a sample in the Methods (see line 258, which says predation was measured but does not say how it was measured).
Response: Thank you, statement changed in Methods to clarify how % predations were recorded: “Mean numbers of MFC/leaf, % parasitism by M. macadamiae, and % predations were recorded from leaves and petioles (n = 30) of infested leaves. Parasitism and predation rates were determined by shape of parasitoid exit holes or predation chewing holes”.
Discussion
Line 351 change “the Hawaii Island” to “the island of Hawaii” Do the same in all places
Response: Thank you, I made the change to island of Hawaii in all places in the text.
Lines 398-399 This is an incomplete sentence, grammatically.
Response: Thank you, sentence changed for clarity to “Our laboratory studies show that MFC parasitism rate by M. macadamiae can range 11– 62% parasitism (Table 1), higher than estimates from field parasitism in Australia (21 – 33% parasitism).
Reviewer 3 Report
Yalemar et al. report the results of laboratory studies on a recently described encyrtid parasitoid of a scale pest of macadamia in Hawaii. The studies have relevance to the release of this wasp for classical biological control. The study would be of interest to readers of Insects, but requires major improvement before it would be suitable for publication.
The authors should tone down their assertions regarding the host specificity of the wasp. They tested 12 species of coccids and Lepidoptera and observed no non-target host feeding or parasitism. I guess that the number of species present in Hawaii is considerably higher than 12, so more caution is merited regarding the potential adverse effects of a field release of this wasp.
The Summary/Abstract, Introduction and Discussion need to be more tightly focused on the aims, results and implications of the study.
The parasitoid had a fairly mediocre performance in the laboratory (6 – 60% parasitism) and is unlikely to contribute greatly to biocontrol in the field, although even moderate population suppression is better than no biocontrol at all.
I have written suggestions and numbered points on a scanned copy of the manuscript.
Numbered points (see scanned copy).
1. Spell out NSW or delete (just say Australia).
2. 50% of the simple summary is preamble. Focus on your results and their implications.
3. This is a very strong statement given the mediocre performance of the parasitoid in laboratory conditions. Please be more cautious.
4. Please use SI units throughout the manuscript (= hectares).
5. Again 50% of Abstract is preamble. Focus on results.
6. Do not repeat words from the title in the keywords.
7. The impact of the pest can be summarized in a couple of lines. Too verbose.
8. Clarify, upper arrow indicates emergence hole, lower arrow indicates predator damage.
9. Are you saying that you don't agree with the reclassification of A. coccineus (from Eriococcus) by Miller & Gimpel 2000? and you are retaining the use of the Eriococcus genus? Please explain why.
10. Better to indicate the total number of non-target insects tested.
11. Five coccinellids? Do you mean five species of coccinellids?
12. NHM, London is repeated.
13. You state in the Methods that the daytime temperature was 34 ± 2 °C (L.175) but in the results you say it was 30-31 °C. Please clarify.
14. Looks like you employed Welch's t-test for unequal variances? Not mentioned in section 2.7 (statistical analysis).
15. This is definitely an overstatement. The wasp did not parasitize any of the 12 species tested. I suspect that the biodiversity of Hawaii exceeds 12 species, so as you did not test ALL possible host species available you cannot be certain that you proved host specificity.
16. Please indicate what Fig 5A and fig 5B show.
17. Please indicate what Fig 6C shows. What do the letters above columns indicate?
18. I don't think that Encarsia is a white wasp of about 1 mm. Please reword.
19. I was unable to understand this text. Please reword.
20. First paragraph of the Discussion – you have already stated that the scale is a pest in the Introduction. Delete.
21. Second paragraph. Presumably certain insecticides are compatible with natural enemies, others less so. This could be moved to the end of the Discussion and shortened considerably.
22. Third paragraph. Again, not of primary concern or the focus of your study. Move to end and summarize.
23. Is Encarsia lounsburyi an autoparasitoid or heteronomous hyperparasitoid? Is it likely to hyperparasitize M. macadamiae for male production? How might this influence the likelihood of successful establishment or the levels of pest control achieved?
24. Your laboratory results indicate that M. macadamiae is a mediocre natural enemy against MFC. Hochberg et al. (1993; doi: 10.1126/science.262.5138.1429)argue that this is likely due to the pest occupying a refuge against parasitoid attack thus reducing the probability of successful classical biocontrol. You should consider this in your discussion.
25. You have no evidence for this assertion. Reword.
26. Is this text particularly relevant to your results? If so, please summarize.
27. Ditto – the situation of the pest in Kenya, China etc. seems tangential to the Results of your study. Suggest you delete this paragraph.
28. Already stated this info. Delete paragraph.
29. Does evidence for monophagy in this parasitoid mainly come from Australia? As this species was only described a few years ago, I suspect that there has been no time to perform systematic analysis of host range in the region of origin. If so, you should be more cautious regarding it's host relations and feeding habits.
Other points. It appears that the authors are not native English speakers, so they should ask an English speaking colleague to review their manuscript prior to resubmission.

Grammatical errors and incomplete sentences.
Author Response
Many thanks for this anonymous reviewer for the efforts to improve our manuscript. All suggested changes were considered in the revised manuscript. I still have to upload the final version after an English-speaking colleague reads it.
Comments and Suggestions for Authors
Yalemar et al. report the results of laboratory studies on a recently described encyrtid parasitoid of a scale pest of macadamia in Hawaii. The studies have relevance to the release of this wasp for classical biological control. The study would be of interest to readers of Insects, but requires major improvement before it would be suitable for publication.
The authors should tone down their assertions regarding the host specificity of the wasp. They tested 12 species of coccids and Lepidoptera and observed no non-target host feeding or parasitism. I guess that the number of species present in Hawaii is considerably higher than 12, so more caution is merited regarding the potential adverse effects of a field release of this wasp.
Response: Thank you so much for reading our manuscript and adding your notes and concerns to improve this article. The number of species of nontargets in Hawaii are mostly adventives. The endemic species were adequately represented in the host specificity studies.
The Summary/Abstract, Introduction and Discussion need to be more tightly focused on the aims, results and implications of the study.
Response: In the revised manuscript we removed some statements from summary and abstract to focus on aims and results of study.
The parasitoid had a fairly mediocre performance in the laboratory (6 – 60% parasitism) and is unlikely to contribute greatly to biocontrol in the field, although even moderate population suppression is better than no biocontrol at all.
Response: This parasitoid is the only parasitoid collected from the native region of MFC and seems to reduce infestation of MFC to a minor pest in NSW Australia. Population of MFC and parasitoid on the island of Hawaii will be monitored after the release to measure the effect of biocontrol. Usually scale insects are good targets for successful biocontrol projects.
I have written suggestions and numbered points on a scanned copy of the manuscript.
Response: Thank you very much for your time reading and suggested notes on the scanned copy of manuscript. All suggested edits are incorporated into the revised version of manuscript.
Numbered points (see scanned copy).
- Spell out NSW or delete (just say Australia).
Response: Thank you we spelled out NSW in the revised manuscript.
- 50% of the simple summary is preamble. Focus on your results and their implications.
Response: We removed some introductory statements from the simple summary (section 2) as suggested.
- This is a very strong statement given the mediocre performance of the parasitoid in laboratory conditions. Please be more cautious.
Thank you for that note, from our host testing this parasitoid is specific to MFC. All other non-targets hosts are not endemic to Hawaii and are adventive pests on many plants. The MFC in Australia, the native region, is a minor pest and this parasitoid is the dominant biotic factor in those native lands.
- Please use SI units throughout the manuscript (= hectares).
Thank you, Acre values changed to Hectares.
- Again 50% of Abstract is preamble. Focus on results.
Response: Thank you, in the revised manuscript we removed some sentences from Abstract to focus on aims and results of study.
- Do not repeat words from the title in the keywords.
Response: Thank you, repeated words replaced in revised manuscript.
- The impact of the pest can be summarized in a couple of lines. Too verbose.
Response: Rampling section on MFC impact now summarized in the new revised manuscript.
- Clarify, upper arrow indicates emergence hole, lower arrow indicates predator damage.
Response: Thank you, these changes added to Figure 3.
- Are you saying that you don't agree with the reclassification of A. coccineus (from Eriococcus) by Miller & Gimpel 2000? and you are retaining the use of the Eriococcus genus? Please explain why.
Response: Sentence corrected for clarity. I agree with Miller & Gimpel 2000 reclassification. The second genus listed in checklist of Eriococcidae, Eriococcus has only one species, Eriococcus coccineus (Cockerell) now moved to Acanthococcus coccineus (Cockerell) new name.
- Better to indicate the total number of non-target insects tested.
Response: Thank you, the table in revised manuscript will have the total number of non-target insect tested instead of the means.
- Five coccinellids? Do you mean five species of coccinellids?
Response: Thank you for clarification. Changed in the revised manuscript to five species of coccinellids.
- NHM, London is repeated.
Response: Thank you, removed repeated NHM, London from the revised text.
- You state in the Methods that the daytime temperature was 34 ± 2 °C (L.175) but in the results you say it was 30-31 °C. Please clarify.
Response: Thank you, this was the average temperature in Honolulu outdoors in summer. I realized that we should be using the indoor Quarantine temperature. This will be changed to indoor quarantine temperature where the study was performed.
- Looks like you employed Welch's t-test for unequal variances? Not mentioned in section 2.7 (statistical analysis).
Response: Thank you, statement added to statistical analysis section (an unequal variances Welch's t-test at P=0.05 level).
- This is definitely an overstatement. The wasp did not parasitize any of the 12 species tested. I suspect that the biodiversity of Hawaii exceeds 12 species, so as you did not test ALL possible host species available you cannot be certain that you proved host specificity.
Response: It would be unrealistic to test all nontarget pests in Hawaii. It will not make sense to make a long list of tested non-targets just to add more negative impact to the table of host specificity. The rest of the species are not endemic to Hawaii and considered pests if the parasitoid can attack them. In past programs, Hawaii managed to release all Aphidiinae (Hymenoptera: Braconidae) parasitoids of aphids without host specificity testing because Hawaii does not have any native aphid to protect.
- Please indicate what Fig 5A and fig 5B show.
Response: Thank you, notes on Figure 5 now revised for clarity.
- Please indicate what Fig 6C shows. What do the letters above columns indicate?
Response: Thank you, notes on Figure 6C now revised for clarity.
- I don't think that Encarsia is a white wasp of about 1 mm. Please reword.
Response: Thank you statement reworded for clarity (Encarsia lounsburyi (Hymenoptera: Aphelinidae) on MFC male scale. White male scale is about 1.0 mm long).
- I was unable to understand this text. Please reword.
Response: Reworded for clarity in revised text “The Pahala site had no significant differences between the densities of E. lounsburyi and C. coeruleus whereas Honomalino had mean count differences of 25 parasitoids to 5 predators individuals/trap/month”.
- First paragraph of the Discussion – you have already stated that the scale is a pest in the Introduction. Delete.
Response: deleted.
- Second paragraph. Presumably certain insecticides are compatible with natural enemies, others less so. This could be moved to the end of the Discussion and shortened considerably.
Response: moved to end of discussion and reduced.
- Third paragraph. Again, not of primary concern or the focus of your study. Move to end and summarize.
Response: moved to end of discussion and reduced.
- Is Encarsia lounsburyi an autoparasitoid or heteronomous hyperparasitoid? Is it likely to hyperparasitize M. macadamiae for male production? How might this influence the likelihood of successful establishment or the levels of pest control achieved?
Response: we do not know until we release and evaluate M. macadamiae in the field. Usually, Encarsia lounsburyi reported attacking males of MFC while M. macadamiae attack females.
- Your laboratory results indicate that M. macadamiae is a mediocre natural enemy against MFC. Hochberg et al. (1993; doi: 10.1126/science.262.5138.1429) argue that this is likely due to the pest occupying a refuge against parasitoid attack thus reducing the probability of successful classical biocontrol. You should consider this in your discussion.
Response: I agree with refugial theory by Hockberg. Our parasitoid shows reduced parasitism on petioles, probably stems. We leave this to actual data and rates of parasitism after the release. Which may be determined by University of Hawaii students. HDOA are concerned with the specificity of biocontrol release. We kept the release of parasitoid until establishment in the fields. Not enough staff and funds to conduct such work in HDOA.
- You have no evidence for this assertion. Reword.
Response: Reworded to “The addition of M. macadamiae to the assembly of natural enemies exploiting MFC will hopefully decrease pest population levels so densities of MFC are reduced to tolerable levels as in the native Australia.
- Is this text particularly relevant to your results? If so, please summarize.
Response: Yes, relevance to show what was released in Hawaii and still save “No non-target parasitism of native insects in Hawaii was recorded by Metaphycus species since their first introduction in 1934”.
- Ditto – the situation of the pest in Kenya, China etc. seems tangential to the Results of your study. Suggest you delete this paragraph.
Response: Deleted most of paragraph. Retained information on South Africa since they are interested in our colony for introduction to their country.
- Already stated this info. Delete paragraph.
Response: Okay, I deleted the paragraph.
- Does evidence for monophagy in this parasitoid mainly come from Australia? As this species was only described a few years ago, I suspect that there has been no time to perform systematic analysis of host range in the region of origin. If so, you should be more cautious regarding it's host relations and feeding habits.
Response: We concur its host specificity in Hawaii using our laboratory studies with endemic and other important groups of insects reported to be attacked by the Metaphycus genus. From the study we conclude the parasitoid is monophagous. Others who would like to use this natural enemy should not rely on our results and should conduct their own testing as we mentioned for the South African.
Other points. It appears that the authors are not native English speakers, so they should ask an English speaking colleague to review their manuscript prior to resubmission.
Response: Thanks for the many edits you made on the manuscript. A native English-speaking colleague will read our revised manuscript before submission.
References will be formatted for INSECTS.
Round 2
Reviewer 3 Report
The authors have taken into account my suggestions. The manuscript is suitable for publication in my opinion.
Minor editing required (during journal production)
Author Response
Thank you so much for reading or manuscript for the second round revision. With your correction and editors' notes the manuscript is greatly improved,